# Splenic Elemental Composition of Breast Cancer-Suffering Rats Supplemented with Pomegranate Seed Oil and Bitter Melon Extract

**DOI:** 10.3390/molecules29091942

**Published:** 2024-04-24

**Authors:** Małgorzata Białek, Tomasz Lepionka, Wiktoria Wojtak, Anna Ruszczyńska, Ewa Bulska, Marian Czauderna, Agnieszka Białek

**Affiliations:** 1The Kielanowski Institute of Animal Physiology and Nutrition, Polish Academy of Sciences, Instytucka 3, 05-110 Jabłonna, Poland; w.wojtak@ifzz.pl (W.W.); m.czauderna@ifzz.pl (M.C.); a.bialek@ifzz.pl (A.B.); 2The Biological Threats Identification and Countermeasure Center of the General Karol Kaczkowski Military Institute of Hygiene and Epidemiology, Lubelska 4 St., 24-100 Puławy, Poland; tomasz.lepionka@wihe.pl; 3Biological and Chemical Research Centre, Faculty of Chemistry, University of Warsaw, Żwirki i Wigury 101, 02-089 Warsaw, Poland; aruszcz@chem.uw.edu.pl (A.R.); ebulska@chem.uw.edu.pl (E.B.); 4School of Health and Medical Sciences, University of Economics and Human Sciences in Warsaw, Okopowa 59, 01-043 Warsaw, Poland

**Keywords:** spleen, rats, breast cancer, elements, pomegranate seed oil, bitter melon extract

## Abstract

The aim of this study was to investigate how dietary modifications with pomegranate seed oil (PSO) and bitter melon aqueous extract (BME) affect mineral content in the spleen of rats both under normal physiological conditions and with coexisting mammary tumorigenesis. The diet of Sprague-Dawley female rats was supplemented either with PSO or with BME, or with a combination for 21 weeks. A chemical carcinogen (7,12-dimethylbenz[a]anthracene) was applied intragastrically to induce mammary tumors. In the spleen of rats, the selected elements were determined with a quadrupole mass spectrometer with inductively coupled plasma ionization (ICP-MS). ANOVA was used to evaluate differences in elemental composition among experimental groups. Multivariate statistical methods were used to discover whether some subtle dependencies exist between experimental factors and thus influence the element content. Experimental factors affected the splenic levels of macroelements, except for potassium. Both diet modification and the cancerogenic process resulted in significant changes in the content of Fe, Se, Co, Cr, Ni, Al, Sr, Pb, Cd, B, and Tl in rat spleen. Chemometric analysis revealed the greatest impact of the ongoing carcinogenic process on the mineral composition of the spleen. The obtained results may contribute to a better understanding of peripheral immune organ functioning, especially during the neoplastic process, and thus may help develop anticancer prevention and treatment strategies.

## 1. Introduction

The spleen is a dark red to blue-black organ and the largest lymphatic internal organ of the body, constituting an important part of the reticuloendothelial system (RES), also known as the mononuclear phagocyte system. It is located in the abdomen, directly beneath the diaphragm, and is connected to the stomach. Its functions include phagocytosis of damaged or old cells and particulate matter, iron recycling, involvement in immunologic response, interception of blood elements, and erythropoiesis [1,2]. Its unique structure corresponds to its functions. The white pulp acts as an immune organ and is a site of production and maturation of B cells and T cells. It regulates inflammation and response to infections. The red pulp, which acts as a phagocytic organ, filters the blood; removes old, damaged, or defective red blood cells (RBCs) and antibody-coated blood cells and bacteria; and serves as a reservoir for blood elements, especially white blood cells (WBCs) and platelets. It also harvests the iron from the old RBCs for recycling into new ones, and it has the ability to produce new RBCs if the bone marrow does not work properly. Splenic macrophages together with hepatic macrophages provide storage for the iron obtained from RBC degradation and are able to release it in response to stimuli from the bone marrow. The spleen is also a place of organization of antibody production against numerous antigens due to the cooperation of T lymphocytes, antigen-presenting cells, and B lymphocytes [1,3,4,5].

Impaired spleen functions are associated with the appearance of different diseases, including sickle cell anemia and malaria. Splenectomy (removal of the spleen) is sometimes required to treat cancers involving the spleen. Metastasis to the spleen is very rare and occurs at the terminal stage of cancer, accompanying multiorgan metastasis and dissemination. Breast, lung, ovary, colorectal, kidney, and skin are the most frequent primary cancers that metastasize to the spleen [6]. Splenomegaly is considered to be connected with poor prognosis in different types of leukemia [7]. Hypersplenism developing after abdominal radiotherapy in gastric carcinoma resulted later in numerous complications [8]. It was also observed that splenectomies correlated directly to a significant increase in malignant tumor induction, as splenectomized rats and mice showed a significant increase in malignant tumor induction [9]. Epidemiological studies by Kristinsson et al. revealed an increased risk of numerous solid tumors (lung, buccal, esophagus, pancreas, liver, colon, and prostate) as well as hematologic malignancies in splenectomized patients [10]. Also, Noma et al. postulated the involvement of the spleen in immunosuppression in advanced gastric cancer patients, which may favor splenectomy in them [11]. 

On the other hand, tumor-associated macrophages and tumor-associated neutrophils, which infiltrate most solid cancers in humans and can promote cancer cell proliferation, metastatic spread, and angiogenesis via different mechanisms, may partially originate from the spleen and splenectomy before or after tumor initiation, and may effectively delay tumor growth [12]. Also, after splenectomy, Stoth et al. observed a reduced number of hematogenous breast cancer lung metastases, which may have resulted from changes in the immune composition of the pre-metastatic niche in the lungs of breast cancer-bearing mice. They concluded that spleen removal was responsible for the changes in the immune microenvironment not only of primary tumors but also of pre-metastatic and metastatic sites [13]. 

Ongoing tumors are known for metabolic reprogramming, which is associated with cellular proliferation, energy storage, and the generation of signaling molecules. We have revealed previously that ongoing neoplastic processes indicate great dysregulation of lipid metabolism, which concerns all classes of lipids and most pathways of their transformation, with a special emphasis on lipid peroxidation and LOX-mediated metabolism [14]. Demand for micronutrients (vitamins and minerals) that are required for physiological cellular processes may also be changed and reprogrammed by the ongoing neoplastic process. The involvement of minerals in oncogenesis has been confirmed, as they are essential to the energetic balance of the cell and its oxidative state. They have a role in DNA stability, epigenetic regulation, and immunological response, and they may also be used as biomarkers. For example, iron from ferritin can react with water to form hydroxyl radical, which is one of the reactive oxygen species (ROS) interacting with lipid bilayers, causing lipid peroxidation; damage of the DNA, proteins, and DNA-repair enzymes; and activation of p53 and NF-κB, resulting in the cell cycle altering. Iron is also present in the first three mitochondrial complexes. Copper is a component of mitochondrial complex IV. Iodide is involved in ROS inactivation. Zinc acting via the zinc protein ERp72 can modulate DNA repair processes and regulate NF-κB. Zinc-containing protein ATOX1 can also modulate expression of cyclin D1 and SOD3 genes, which are involved in the regulation of the oxidative environment. Moreover, iron and zinc, being the components of the DNA replication complex, are involved in DNA replication [15,16]. The cellular levels of elements are regulated, i.e., by dietary intake, physiological or pathological state, dietary modifications (especially vegetable and fruit consumption as rich dietary sources of vitamins and minerals), and dietary supplements. According to Velicer et al., many cancer patients (14–32%) decided to initiate dietary supplementation after diagnosis; the highest number of dietary supplement usage was noted among breast cancer patients [17]. However, a systematic review of Huang et al. did not confirm the presence or absence of benefits resulting from vitamins and mineral dietary supplements in preventing cancer occurrence [18]. Moreover, the fact that most cancer patients who decide to use dietary supplements neither consult with their physicians, pharmacists, or dietitians about this decision nor inform their caregivers is alarming [17]. As we indicated previously, the effect of dietary supplements (pomegranate seed oil (PSO) and bitter melon aqueous extract (BME)) depends on the state of an organism and can be modified or even reversed in pathological conditions [14,19,20,21,22]. 

Pomegranate seeds contain from 7.6 to 16.2% of PSO and are an abundant source of phospholipids, phytosterols, tocopherols, tocotrienols, squalene, and triterpene. Also, PSO is known as a rich source of *cis*9*trans*11*cis*13C18:3 fatty acid (punicic acid; the C18:3 isomer). Bitter melon fruits are sources of cucurbitane-type triterpenoids and glycosides, phenolic acids, flavonoids, carotenoids, essential oils, fatty acids, sterols, saponins, some proteins and amino acids, micro- and macro-elements, and vitamins [23].

Therefore, the aim of the present study was to investigate the influence of diet modifications with PSO and BME on the content of selected elements in the spleen of rats in the physiological state and with co-existing mammary tumorigenesis.

## 2. Results

### 2.1. Element Content in Fodder and Dietary Supplements

The content of macro- and microelements in fodder, which was consumed by the animals daily ad libitum, as well as in the applied dietary supplements, is presented in Table 1. In Labofeed H, the most abundant macroelements are K, Ca, Mg, and Na, which make them predominant in the diets of experimental animals. In dietary supplements, the levels of elements were lower compared to the laboratory fodder. In PSO, K and Fe were detected in the highest amounts, whereas the contents of Na, Cu, Co, Al, Sr, Pb, Cd, B, and Tl were below the LOD. In the dry matter of bitter melon fruit, all elements were determined, and in the BME given to animals daily as drinking fluid, only Se and Cd were absent. Taking into account the content of the examined elements in applied dietary supplements and the amounts of consumed supplements, it seems that diet modification with PSO or/and BME did not influence the mean dietary intake of the investigated elements in all experimental groups, as the part of the daily dose of elements obtained from PSO or/and BME was negligibly small. 

### 2.2. Element Content in Spleen Samples

The performed ICP-MS analyses enabled the determination of 4 macroelements (K, Mg, Na, and Ca) and 14 microelements in spleen samples (Table 2). In the case of sodium, the highest levels were determined for Gplus and CONplus, which differed significantly from the levels determined in the M group. Among macroelements, no significant differences among experimental groups were observed, except for potassium levels.

Among all microelements, Fe was the most abundant. Its highest contents were determined in CON, CONplus, and Mplus, whereas the lowest amounts were detected in Gplus and GMplus, which may indicate the influence of PSO supplementation on iron levels and metabolism in the spleen. For Zn, Cu, and Mn, no significant differences were observed among experimental groups. In the case of Se, Co, Cr, and Ni, the levels of these elements were higher in groups of animals that were not subjected to DMBA treatment than in DMBA-treated groups. Carcinogen application also seemed to influence Al, Sr, B, and Tl levels in the spleen, as the contents of these elements were higher in DMBA-treated groups than in animals not subjected to carcinogen treatment. The highest levels of Pb were determined in the CON group, and they were significantly higher than in the Mplus and GMplus groups. The CON group was also characterized by the highest levels of Cd, which significantly exceeded the amounts of this element detected in the spleen of the Mplus, Gplus, and GMplus groups. Also, the G group was characterized by high levels of Cd, which significantly differed from amounts of this element detected in Gplus and GMplus.

### 2.3. Cluster Analysis

The results of CA are presented as dendrograms in Figure 1 and Figure 2. The application of the less restrictive Sneath’s criterion (66%) to the dendrogram analysis allowed for three clusters to be distinguished that differentiated the examined elements (Figure 1). The first cluster (Cl1) included three elements: Mg, Zn, and K; the second cluster (Cl2) included ten elements: Ca, Fe, Cu, Mn, Se, Co, Cr, Ni Pb, and Cd; and the third cluster (Cl3) included five elements: Na, Al, B, Sr, and Tl. The application of the less restrictive Sneath’s criterion (66%) to the second dendrogram analysis allowed for three clusters to be distinguished that differentiated the examined spleen samples (Figure 2). 

The first cluster (Cl1) included most of the spleen samples obtained from animals treated with DMBA, the second cluster (Cl2) included most of the spleen samples obtained from animals not treated with the carcinogen, and the third cluster (Cl3) was a mixture and included three spleen samples from the M group, one sample from the G group, three samples from the GM group, two samples from the CONplus group, six samples from the Mplus group, five samples from the Gplus group, and seven samples from the GMplus group. Similarity analysis was performed by grouping features and objects to prepare a heat map (Figure 3). 

Statistical analysis revealed significant differences in the content of most examined elements among existing clusters of spleen samples (Table 3). Only Mg and Na content did not differ among clusters. Cluster Cl1 was characterized by the highest contents of Al, Sr, B, and Tl, which significantly exceeded the content of these elements in clusters Cl2 and Cl3. Cluster Cl2 was characterized by the highest content of Zn, which was significantly higher than its content in Cl3, and by the highest contents of Se, Co, Ni, and Pb, which were significantly higher than in clusters Cl1 and Cl3. For cluster Cl3, the lowest contents of K, Ca, Cu, Mn, Se, Cr, Pb, and Cd were observed, which were significantly lower than in Cl2 (for K) or in Cl1 and Cl2 (for the rest of the elements) (Table 3). 

### 2.4. Principal Component Analysis (PCA)

The contents of all assayed elements were subjected to PCA. As shown by the screen test, three principal components (PCs) were enough to explain 64.7% of the total variance (Table 4). 

The highest contribution (>70%) to the first PC exhibited the content of Cd, Se, Cr, Mn, Pb, Cu. The levels of B, Al, Co, Sr, and Tl demonstrated variance in the second PC. The third PC gained variance (>60%) from the content of Mg, Zn, and K. The PC1/PC2, PC1/PC3, and PC2/PC3 biplots enabled the samples from the DMBA-treated and untreated animals to be divided; however, no clear separation among groups receiving various supplementation was observed (Figure 4). Strong positive correlations were observed between contents of B and Al, and Sr and Tl, as well as among Cu, Cd, Mn, Cr, and Pb, while a negative correlation was observed between Na, Mg, and Ca levels.

### 2.5. Linear Discriminant Analysis (LDA)

LDA was used to obtain appropriate classification rules for the examined samples of spleen. Relevant discriminant functions were calculated in a stepwise progressive method. In the performed analysis, 16 variables were included in the final model, from which 10 were statistically significant. The applied canonical analysis allowed for seven discriminant functions to be distinguished, from which four discriminant functions (DF1–DF4) were statistically significant. DF1 was the most significant function, as it explained over 93.63% of the discriminatory power. DF2, DF3, and DF4 explained only 2.40%, 1.76%, and 1.16% of the discriminatory power, respectively (Table 5).

Analysis of the canonical mean variables indicated that DF1 had the greatest impact on the distinction of groups of animals treated with DMBA (“plus” groups) compared to groups that were not subjected to carcinogen treatment. Overall, a good separation of experimental groups regarding the presence of chemical carcinogens was obtained. 

The calculated classification matrix indicated that the average classification efficiency based on the calculated functions was 84.44% (Table 6). For individual groups, these coefficients were as follows: 100% for M, 91.67% for Mplus, 90.91% for CONplus and GMplus, 83.33% for CON and GM, and 75.00% for Gplus. The lowest coefficient (58.33%) was revealed for the G group.

## 3. Discussion

The immune system is one of the most complex systems in the body [24], as it consists of internal organs and tissues, cellular elements, and humoral products.

The spleen works together with the liver and bone marrow to activate the defense response through innate and adaptive immunity [25]. Breast cancer has been considered one with the least immune response [24]. Progression through the carcinogenic pathway from normal breast tissue to breast cancer is accompanied by quantitative and qualitative changes in the nature and location of the immune cell population [26,27]. These cells generate an anti-cancer immune response, which is, however, inhibited by the action of myeloid-derived suppressor cells, which becoming functionally active and extramedullary in organs such as the spleen [27]. The coordination of immunity across organs is fundamental to cancer development and progression. However, whether metabolic alterations in secondary lymphoid tissues (e.g., a spleen) may affect anti-tumor immunity is still under consideration [28]. 

As discussed previously, there were no spontaneous tumors of any kind in animals not treated with DMBA, whereas mammary cancer incidence in “plus” groups was 42% in CONplus, 67% in Mplus, and 100% in Gplus and GMplus groups, respectively [22]. This may be since DMBA-treated animals have decreased immune response, in terms of, e.g., antibody production, cell-mediated cytolysis, and number of T cells [29,30]. One hundred percent morbidity in groups receiving PSO together with the fact that the first mammary tumors appeared much earlier in Gplus than in other supplemented groups indicated PSO’s ability to accelerate the neoplastic process in this model. 

The effect of dietary fatty acids (FAs) on the immune system was suggested by early epidemiological studies of the incidence of multiple sclerosis. It was also confirmed both in vitro and ex vivo that polyunsaturated fatty acids (PUFAs) may influence such essential lymphocyte functions as proliferation, activity of cytotoxic T-lymphocyte and NK cells, production of interleukin 2 (IL-2) and prostaglandin E2 (PGE2), and presentation of antigens. It should be emphasized that this influence depends on the FA type, e.g., dietary sources of n-3 PUFAs exhibit immunosuppressive properties, while oils rich in n-6 PUFAs are less suppressive, and oleic acid—a monounsaturated fatty acid (MUFA) representative—acts as an immunomodulator [31,32,33]. Also, two main isomers of conjugated linoleic acid (c9t11 CLA and t10c12 CLA) enhance IgA, IgG, and IgM production in spleen lymphocytes and mesenteric lymph nodes of rats [34]. Our previous results confirmed that dietary sources of isomers of conjugated linolenic acid (CLnA) also affect the FA profile of the spleen in both physiological and pathological states [35]. Thus, as a continuation and an extension, we also decided to evaluate the effect of such experimental conditions on the elemental composition of a rat’s spleen. 

One of the basic indicators of immune system functioning may be considered the relative weight of lymphoid organs, with the following correlation: The greater the weight of these organs the stronger their immune activity [36]. In the present study, neither applied supplementation nor carcinogen treatment influenced the absolute spleen weight. However, in the case of relative spleen weight, a significant increase was observed in all DMBA-treated groups, despite the applied supplementation [22]. This may be explained by the fact that spleen volume increased as a function of tumor growth [30]. There are many types of cells located in the enlarged spleen of tumor-bearing organisms, and they might have various effects on tumor progression via different mechanisms [37]. 

Nowadays, tumor and non-tumoral lymphoid tissues (like bone marrow and spleen) and non-invasive measurements of glucose metabolism are used as tumor aggressiveness and cell proliferation indicators as well as to predict the recurrence and onset of the metastasis of a broad spectrum of tumor types [38]. The glucose metabolism seems to be of utmost importance, as the spleens of mice with breast cancer are metabolically reprogrammed to the glycolytic state. Magnesium, by influencing the activity of insulin receptor, tyrosine kinase, and glucose transport protein 4 (GLUT4), directly participates in the regulation of glucose translocation into the cell. Intracellular transformation of glucose into adenosine-5′-triphosphate (ATP) also affects the equilibrium between K^+^ and Ca^2+^ ion flow through the membrane [28]. Although in the current study the potassium levels in the spleen of rats were unaffected by any experimental factor, the contents of Mg and Ca significantly differed among groups (Table 2). It may be observed that in terms of the neoplastic process, BME strongly affected the Mg levels in the spleen, as when applied separately, both supplements showed an antagonistic effect, whereas when supplemented jointly, the effect of PSO seemed to be less pronounced/neutralized by BME. Where Ca is concerned, a different effect of the experimental factors was observed. In DMBA-receiving rats, both supplements acted synergistically and decreasingly, which was confirmed by the lowest levels of Ca in the spleen of animals from the GMplus group. The lowering effect of BME and PSO on splenic levels of Ca was observed in DMBA-deprived rats, but there was a lack of interaction between supplements. 

Microelements play different important roles in many metabolic processes in all biological systems in both physiological and pathological conditions [39]. Immune cells and their effector activities are sensitive to marginal and moderate deficiencies of trace elements, e.g., zinc deficiency in mice decreased levels of IgA both in serum and in mucosa [40]. An adequate supply of trace elements is required for the structure and function of metalloproteins that participate in energy production and protection against oxidative stress. An element strongly involved in antioxidant processes is selenium. Se exerts its biological functions through selenoproteins [41], among which more than half play a key role in neutralizing free radicals. Levels of Se were significantly affected by both experimental factors. The decrease in the splenic selenium content of rats exposed to cancerogenic agent seems to confirm that the neoplastic process intensifies free radical activity. 

Trace element status also affects the synthesis and secretion of cytokines and chemokines that modulate the activities of immune cells and other cells [42]. Indeed, the spleen, as an internal organ with high metabolic activity, contains a reservoir of myeloid progenitors and monocytes, constituting an important source of inflammatory cells [43]. CLnA isomers, punicic acid (c9t11c13C18:3, PA), and α-eleostearic acid (c9t11t13C18:3, ESA) found in dietary supplements applied to laboratory animals in the present study exhibit anti-inflammatory properties, as claimed by Saha et al. [44]. 

Currently, the spleen is considered a prominent site of extramedullary hematopoiesis in cancers, during which not only myeloid cells are produced, e.g., in late-stage cancers, the spleen generates unique erythrocytic cell populations [45]. Tumor growth could play a pivotal role in reprogramming the host immune system by regulating hematopoiesis, which might be associated with the expansion of bone marrow- and/or spleen-derived immunosuppressive cells [38]. Tumors interfere with splenic erythropoiesis, resulting in the accumulation of late erythroid precursors [37]. Erythroid cells form rose junctions with invasive pathogens (e.g., tumor cells) and facilitate their elimination as macrophages pass through the liver and spleen. Thus, the immunity mediated by erythrocytes may prevent cancer cells from spreading through the bloodstream [46]. RBCs are hydrolyzed in the phagolysosome of the splenic macrophages. Proteolytic degradation of hemoglobin takes place, and hem is released and further catabolized into biliverdin, carbon monoxide, and ferrous ion (Fe^2+^) [2]. 

Dietary supplementation of healthy animals caused a decrease in the splenic content of Fe to the greatest extent in the groups receiving only PSO. In rats treated with DMBA, the lowest level of this element was observed in the spleen of animals receiving joint supplementation with BME and PSO, whereas the highest was in the Mplus group. Another trace element important for the hematopoietic process is Co, which, as a part of cobalamin (vitamin B12), takes part in the multiplication and division of RBCs. Co is also essential for the activation of enzymes and plays an important role in the functioning of abzymes with DNase and RNase activities. Abzymes are monoclonal antibodies with catalytic properties, which may be found in organisms suffering from autoimmune diseases. Ions of Co^2+^-activated mammalian DNase I, however, the relative activity of metal-dependent enzymes hydrolyzing DNA depends also on the type and timing of the disease as well as on the individual variability [39]. Levels of Co were significantly lower in the spleen of DMBA-treated rats in comparison to healthy animals and remained unchanged, regardless of the applied supplementation. The same dependence was confirmed by the results obtained for the revealed clusters of animals (Table 3). A similar tendency in the fluctuation depending on the experimental factors may be noticed in the content of Ni. Nickel is a transition metal, present in active sites of eight enzymes, e.g., hydrogenase and urease, with similar properties to Co. Levels of this trace element were higher in the spleen of healthy rats in comparison to DMBA-treated animals. When Ni accumulates to a certain level in the body, it can be toxic to living organisms [41]. The International Agency for Research on Cancer (IARC) classifies Ni, Cd, and hexavalent Cr as human carcinogen group 1. BME supplemented alone decreased the content of both Cd and Cr to the greatest extent in the spleens of healthy rats. On the other hand, in DMBA-treated animals, the strongest properties for lowering Cd content exhibited PSO, supplemented alone or jointly with BME, whereas in the case of Cr, it was BME that exhibited such a strong decreasing effect, either alone or in combination with PSO. 

Although significant differences were observed between experimental groups in the content of most of the examined macro- and microelements, all chemometric methods clearly confirmed that the ongoing carcinogenic process had the greatest impact regarding the mineral composition of the spleen. The visible division into DMBA-treated and DMBA-untreated groups was observed in CA, PCA, and LDA. CA and PCA also revealed great similarities in the observed trends. Cl1, containing only animals treated with DMBA, was characterized by the highest amounts of Al, Sr, B, and Tl (Table 3), which was confirmed by the fact that strong positive correlations were observed between the contents of these elements in PCA (Figure 4). There is much evidence that these elements are involved in carcinogenic processes in different organs and tissues (elevated levels in the organisms, increased intake, and the potential to stimulate oxidative stress or weaken the antioxidant defense of the organisms were confirmed) [47,48,49,50,51]. Our results also confirmed that ongoing mammary tumorigenesis was associated with elevated levels of these elements in the spleen. On the other hand, it should be noticed that Cl2, including only DMBA-deprived animals, was characterized by the highest contents of Zn and Se, which are known for their anti-carcinogenic properties [52,53]. This also confirms that the elemental composition of tissues and organs corresponds to the physiological and pathological conditions of the organism and may influence the risk of cancer development.

## 4. Materials and Methods

### 4.1. Dietary Ingredients

#### 4.1.1. Bitter Melon Aqueous Extract (BME)

Bitter melon dried fruits were purchased from the local market in Warsaw, Poland, as the main ingredient of tea for brewing (Tra Kho Qua, Hung Phat Corp, Ho Chi Minh City, Vietnam). Daily fresh extract of 1% (*w*/*v*) was prepared according to the manufacturer’s description. An exact portion of the dried material was weighed, and a matching measured quantity of hot water (80 °C) was added to obtain 1% extract (*w*/*v*). After 10 min, the fresh extract was filtered, brought to room temperature, and administered to the animals ad libitum.

#### 4.1.2. Pomegranate Seed Oil (PSO)

Pomegranate seed oil from the seeds of pomegranate fruits was purchased from the retail market in Warsaw, Poland. It was cold pressed, unrefined, 100% seed oil (ECOSPA), originating from Great Britain. PSO was stored unopened at 8 °C in the manufacturer’s original dark glass package. Before administration to the animals, it was brought to room temperature. PSO was administered to the animals via gavage daily in the amount of 0.15 mL per animal. 

### 4.2. Animal Experiment

The II Local Ethical Committee on Animal Experiments in Warsaw approved the whole experiment as well as the guiding principles for the use and care of laboratory animals (decisions no 56/2013 and 54/2015). The animals were purchased from the Central Laboratory of Experimental Animals, Medical University of Warsaw, Poland. Maiden Sprague-Dawley rats (n = 96, age 30 days) were kept in the animal room in plastic colony cages (3 individuals per cage) at a constant temperature (21 ± 1 °C), with a relative humidity of 50–60% and a 12 h light–dark cycle during the entire experiment. Moreover, during the experiment, they were provided ad libitum access to a standard laboratory fodder Labofeed H (Feed and Concentrates Production Plant, A. Morawski, Żurawia 19, Kcynia, Poland) and drinking liquid. The detailed composition of Labofeed H fodder was published previously [22]. After 1 week of adaptation, the animals were randomly divided into 8 groups of 12 individuals each. Detailed characteristics of the experimental groups are presented below:-CON and CONplus—control groups without diet supplementation, fed a standard diet and tap water ad libitum;-M and Mplus—animals fed a standard diet supplemented with 1% BME ad libitum;-G and Gplus—animals fed a standard diet and water ad libitum and given 0.15 mL/day of PSO via gavage,-GM and GMplus—animals fed the standard diet and supplemented with both 0.15 mL/day of PSO via gavage and 1% BME ad libitum.

Rats of four groups marked with “plus” obtained on the 50th day of life 7,12-dimethylbenz[a]anthracene (DMBA) in the dose of 80 mg/kg body weight for the induction of mammary tumors. The chemical composition of the applied supplements was described previously [23]. Throughout the experiment, the rats were monitored daily for signs of welfare disorders (e.g., appetite loss, ruffling, sluggishness, apathy, hiding, curling up) by a qualified veterinarian. They were also checked for any specific signs of health deterioration and weighed each week. The carcinogenic agent was dissolved either in PSO (for the Gplus and GMplus groups) or rapeseed oil (for the CONplus and Mplus groups) and was administered as a solution via gavage intragastrically. The experimental period lasted 21 weeks, and afterwards, the rats were decapitated and exsanguinated, and their spleens were removed, weighed, and stored at −80 °C for further chemical analysis.

### 4.3. ICP-MS Element Determination 

Analytical-grade reagents, solvents, and standards were obtained from Merck (Darmstadt, Germany), J.T. Baker (Philipsburg, NJ, USA), and PerkinElmer (Waltham, MA, USA). Deionized water obtained from the Milli-Q system (18.2 MΩ; Merck, Darmstadt, Germany) was used for samples and standard dilution. A standard reference material, 1577c bovine liver (NIST, Gaithersburg, MD, USA), was used for verification of the analytical procedure of the determination of selected elements (Fe, K, Mg, Mn, Na, Ca, Cd, C, Cr, Cu, Ni, Pb, Se, Sr, Zn) in animal tissue. To investigate the intricate relationships between dietary supplementation and elemental composition, inductively coupled plasma mass spectrometry (ICP-MS) was employed. Certified reference materials with certified values for the selected elements in surface water (SPS-SW1 and SPS-SW2, Spectrapure Standards, Oslo, Norway) were used to check the accuracy of the measurements by ICP-MS. 

The content of elements in thawed spleens (approximately 50 mg) was determined in tissues subjected to mineral digestion in 65% nitric acid in glass tubes of a microwave ultraWave closed system with Single Reaction Chamber technology (Milestone, Shelton, CT, USA) for 15 min at up to 200 °C and 10 min at 200 °C. After cooling down to room temperature, digests were diluted in deionized water according to the level of the measured element. Due to the limited amount of material, the analytical procedure was conducted for a single weight of test samples. 

In this study, a PerkinElmer Sciex quadrupole spectrometer (NexION 300D, Waltham, MA, USA) was used for isotope-specific detection of selected elements (^11^B, ^23^Na, ^24^Mg, ^25^Mg, ^27^Al, ^39^K, ^42^Ca, ^57^Fe, ^53^Cr, ^55^Mn, ^59^Co, ^60^Ni, ^63^Cu, ^66^Zn, ^88^Sr, ^111^Cd, ^205^Tl, and ^208^Pb). The spectrometer was equipped with a conventional sample introduction system for liquid samples: a quartz cyclonic spray chamber and a Meinhard nebulizer. The working conditions of the spectrometer were as follows: radio frequency plasma power—1350 W, constant nebulizer gas (Ar) flow—0.9 dm^3^/min. Transient signals of the selected isotopes were monitored (1 reading/5 sweeps/3 replicates) with a dwell time of 100 ms/isotope. Quantitation was achieved by 5-point external calibration (concentration range: 1 μg/dm^3^–100 μg/dm^3^). Limits of detection (LODs) and limits of quantitation (LOQs) were calculated for each element by summing the mean of ten measurements of blank samples and three standard deviations for LOD, or ten standard deviations for LOQ, following the IUPAC recommendation [54]. These values are presented in Appendix A. Blank samples were prepared under the same conditions as the samples. The contents of elements in the fodder, PSO, and BME (both in dry matter and in fluid) are given in Table 1. Detailed chemical characteristics of the dietary supplements are presented in Appendix A.

### 4.4. Statistical Analysis

All data were presented as mean values ± standard deviation. For variables with skew distribution, data were transformed into logarithms, retransformed after calculations, and presented as mean and confidence intervals. Statistica 13.5 software (StatSoft, Tulsa, OH, USA) was used for the statistical analysis. To eliminate the outliers, the obtained results were tested with the Q-Dixon test followed by a one-way ANOVA and a RIR Tukey post-hoc test to verify the differences among groups (marked * in Table 2). For variables that were not characterized by the normality of distribution, the obtained results were tested with the Kruskal-Wallis test, which is a non-parametric equivalent of the one-way ANOVA, with a post-hoc multiple-comparisons test. The acceptable level of significance was established as *p* ≤ 0.05.

To verify whether the diet modifications and applied experimental conditions significantly affected group diversity, chemometric analyses were performed. Element content in the spleen samples were used as descriptors to study possible discrimination of the spleen samples. Prior to the analyses, the original data were transformed into natural logarithms and then auto-scaled (standardized).

Cluster analysis (CA) was carried out using the agglomeration method. Euclidean distance was used as the distance determination method, and the Ward method was used as the agglomeration method. The application of the less restrictive Sneath’s criterion (66%) was used for dendrogram analysis and cluster distinguishing (Figure 1 and Figure 2). To determine the differences among existing clusters of spleen samples, a one-way ANOVA with a post hoc RIR Tukey test was performed. The accepted significance level was established as *p* ≤ 0.05. Similarity analysis was performed by grouping features and objects for variables differing significantly among existing clusters to prepare a heat map (Figure 3).

To evaluate whether elemental composition in spleens significantly differs among dietary groups and according to carcinogenic agent usage, principal component analysis (PCA) was performed. A matrix of 20 variables (the content of the detected elements) was used. The number of principal components (PCs) was chosen by the screen test criterion. Results were presented as case projections on the plane of factors (PC1/PC2, PC1/PC3, PC2/PC3) (Figure 4).

Next, in order to obtain appropriate classification rules for spleen samples in the experimental groups, a linear discriminant analysis (LDA) for the examined variables (element content) differing significantly among clusters was performed. Relevant discriminant functions were calculated in a stepwise progressive method, with the adopted tolerance value of 1 − R^2^ = 0.01 to optimize the LDA.

## 5. Conclusions

Understanding the immune system’s functioning, its components, and peripheral organs like the spleen during the neoplastic process is crucial for developing effective anticancer prevention and treatment strategies. This information is also important as a preliminary step to confirm/justify the use of PSO and BME as ingredients in functional foods or pharmaceuticals.

## Figures and Tables

**Figure 1 molecules-29-01942-f001:**
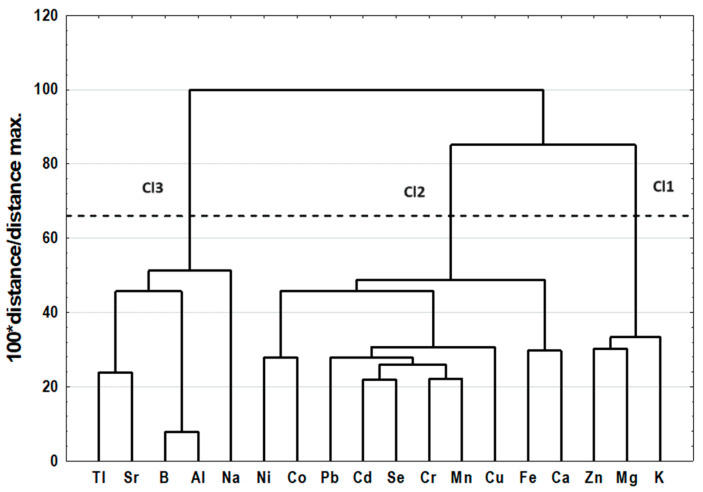
Dendrogram of similarities of examined features (variables).

**Figure 2 molecules-29-01942-f002:**
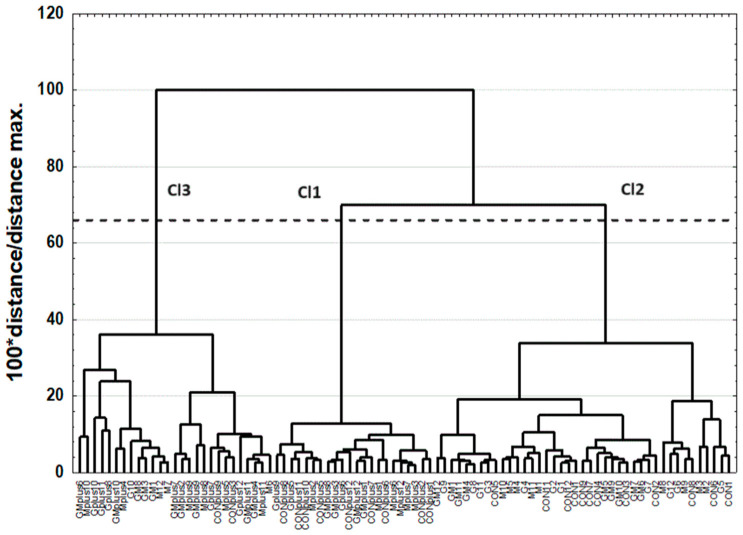
Dendrogram of similarities of examined objects (individuals).

**Figure 3 molecules-29-01942-f003:**
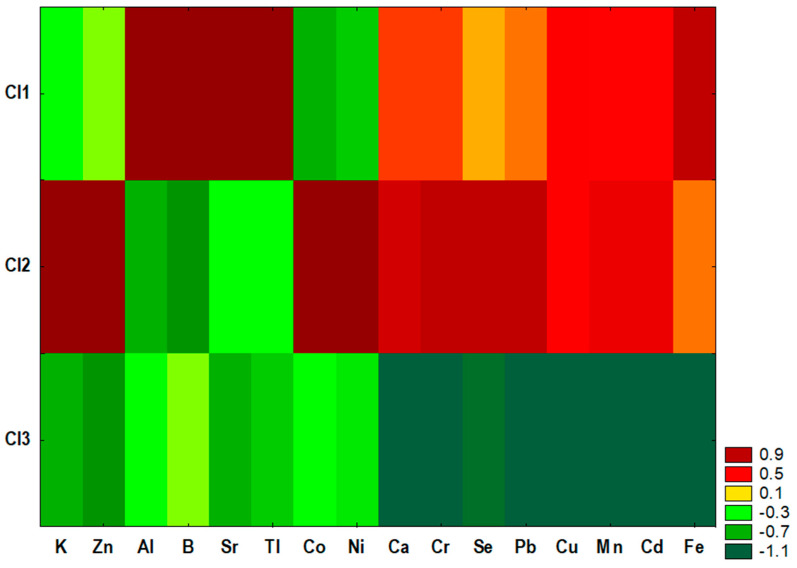
Similarity analysis performed by grouping features and objects to prepare a heat map.

**Figure 4 molecules-29-01942-f004:**
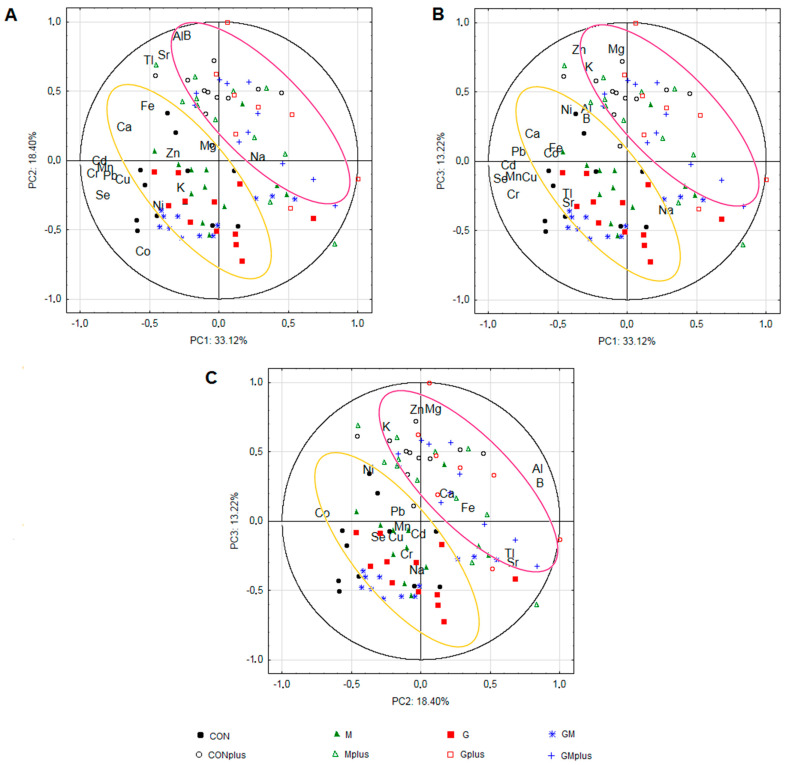
Biplots of mineral content in the spleen of rats from dietary supplementation groups: (**A**) PC1/PC2, (**B**) PC1/PC3, (**C**) PC2/PC3. CON—animals without dietary supplementation; CONplus—animals without dietary supplementation treated with DMBA; M—animals receiving BME; Mplus—animals receiving BME treated with DMBA; G—animals receiving PSO; Gplus—animals receiving PSO treated with DMBA; GM—animals receiving both PSO and BME; GMplus—animals receiving both PSO and BME treated with DMBA.

**Table 1 molecules-29-01942-t001:** Elemental composition of dietary ingredients.

	Labofeed H	PSO	Bitter Melon (Dry Matter)	BME (Fluid)
	mg/kg			µg/L
	Macroelements			
K	18,570	13.66	28,896	524,761
Mg	2930	<LOQ	2913	38,942
Na	2416	<LOD	180	14,018
Ca	7686	<LOD	1896	26,924
	Microelements			
Fe	225.9	4.13	160.3	263.4
Zn	34.28	<LOD	15.61	399.3
Cu	9.88	<LOD	6.79	80.32
Mn	43.35	<LOD	239.8	1741
Se	0.16	0.05	0.11	<LOD
Co	0.23	<LOD	0.16	1.78
Cr	0.29	0.56	1.81	22.18
Ni	1.02	<LOD	2.61	47.25
Al	149.6	<LOD	527.1	3427
Sr	10.74	<LOD	17.0	120.8
Pb	<LOD	<LOD	1.41	6.90
Cd	<LOD	<LOD	0.06	<LOD
B	<LOD	<LOD	<LOQ	212.8
Tl	<LOQ	<LOD	<LOD	<LOD

LOD—limit of detection, LOQ—limit of quantitation.

**Table 2 molecules-29-01942-t002:** Content of elements (per kg of wet matter) in spleens of rats from experimental groups.

	CON	M	G	GM	CONplus	Mplus	Gplus	GMplus	*p*-Value
	Macroelements								
K [mg/kg]	5206 ± 168	5098 ± 277	5102 ± 274	5091 ± 189	4908 ± 587	5024 ± 597	4269 ± 870	5091 ± 273	0.0815
Mg [mg/kg]	255.2 ± 22.9	244.7 ± 18.9	240.2 ± 32.3	232.1 ± 17.2	244.3 ± 26.2	257.3 ± 29.8	223.1 ± 36.5	258.4 ± 20.0	0.0276
Na [mg/kg]	19.53 ± 10.74	11.49 ± 5.87 ^a,b^	16.98 ± 10.48	17.65 ± 5.05	26.07 ± 10.16 ^a^	17.43 ± 11.29	51.60 (11.33–235.0) ^b^	13.85 ± 9.25	0.0048
Ca [mg/kg]	188.2 ± 38.0 ^a,b^	166.9 ± 37.9 ^c^	169.3 ± 30.9 ^d^	176.9 ± 28.5 ^e^	191.3 ± 25.0 ^f,g^	149.9 ± 35.4	128.9 ± 57.2 ^a,g^	115.6 ± 21.5 ^b,c,d,e,f^	<0.0001 *
	Microelements								
Fe [mg/kg]	3939 ± 783 ^a,b^	3476 ± 478	2885 ± 496	3044 ± 155	3665 ± 561 ^c^	3925 ± 1734 ^d,e^	2420 ± 818 ^a,d^	2130 ± 1152 ^b,c,e^	<0.0001
Zn [mg/kg]	22.98 ± 2.71	26.31 ± 7.89	22.91 ± 4.21	21.26 ± 2.09	21.27 ± 2.24	22.19 ± 1.67	20.34 ± 3.71	21.79 ± 0.91	0.2763
Cu [mg/kg]	3.38 ± 0.76	2.82 ± 0.77	2.86 ± 0.64	2.96 ± 0.71	2.98 ± 0.66	2.85 ± 0.49	2.45 ± 0.59	2.84 ± 0.46	0.1581 *
Mn [mg/kg]	0.59 ± 0.11	0.53 ± 0.11	0.52 ± 0.11	0.55 ± 0.15	0.50 ± 0.08	0.51 ± 0.14	0.42 ± 0.10	0.50 ± 0.18	0.1937
Se [mg/kg]	0.73 ± 0.06 ^a,b,c,d^	0.63 ± 0.09	0.67 ± 0.12 ^e^	0.68 ± 0.11 ^f^	0.58 ± 0.10 ^a^	0.57 ± 0.14 ^b^	0.48 ± 0.09 ^c,d,f^	0.54 ± 0.07 ^d^	<0.0001
Co [mg/kg]	0.054 ± 0.010 ^a,b,c,d^	0.046 ± 0.010 ^e,f^	0.059 ± 0.016 ^g,h,i,j^	0.064 ± 0.016 ^k,l,m,n^	0.010 ± 0.001 ^a,e,g,k^	0.010 ± 0.003 ^b,h,l^	0.009 ± 0.002 ^c,f,i,m^	0.010 ± 0.004 ^d,j,n^	<0.0001
Cr [mg/kg]	0.51 ± 0.15 ^a,b^	0.38 ± 0.08	0.46 ± 0.10	0.50 ± 0.10	0.40 ± 0.07	0.36 ± 0.10 ^a^	0.39 ± 0.09	0.36 ± 0.11 ^b^	0.0019*
Ni [mg/kg]	0.16 ± 0.06 ^a,b,c^	0.17 ± 0.14 ^d,e^	0.14 ± 0.12 ^f,g^	0.11 ± 0.08	0.03 ± 0.01 ^a,d,f^	0.05 ± 0.02 ^b^	0.09 ± 0.06	0.03 ± 0.01 ^c,e,g^	<0.0001
Al [mg/kg]	49.33 ± 9.19	52.13 ± 10.33	42.02 ± 7.53 ^a^	41.29 ± 3.22 ^b,c^	57.80 ± 6.41 ^a,b^	51.87 ± 11.71	55.44 ± 16.06	53.82 ± 10.04 ^c^	0.0003
Sr [mg/kg]	0.20 ± 0.04	0.17 ± 0.02 ^a,b^	0.18 ± 0.06 ^c^	0.17 ± 0.04 ^d^	0.28 ± 0.07 ^a,c,d^	0.24 ± 0.07	0.25 ± 0.06 ^b^	0.21 ± 0.06	<0.0001
Pb [mg/kg]	0.18 ± 0.05 ^a,b^	0.16 ± 0.05	0.14 ± 0.03	0.16 ± 0.06	0.14 ± 0.02	0.12 ± 0.03 ^a^	0.13 ± 0.03	0.11 ± 0.03 ^b^	0.0024
Cd [μg/kg]	3.76 ± 0.69 ^a,b,c^	2.77 ± 0.71	2.98 ± 0.74	3.27 ± 0.68 ^d,e^	3.02 ± 0.88	2.62 ± 1.11 ^a^	2.02 ± 0.53 ^b,d^	2.08 ± 0.74 ^c,e^	<0.0001 *
B [mg/kg]	72.34 ± 13.39	77.61 ± 15.63	64.27 ± 12.38 ^a^	63.74 ± 4.96 ^b^	87.17 ± 9.56 ^a,b^	76.22 ± 18.79	82.65 ± 24.57	83.47 ± 17.62	0.0018
Tl [μg/kg]	9.59 ± 2.82	6.88 ± 1.53 ^a,b,c^	8.03 ± 3.14 ^d^	7.46 ± 2.25 ^e,f^	11.99 ± 2.59 ^a,d^	10.85 ± 3.40 ^b,e^	9.37 ± 3.08	11.76 ± 3.99 ^c,f^	<0.0001 *

Data are shown as mean values ± standard deviation (SD). *p*-value ≤ 0.05—significant differences among groups in one-way ANOVA (*) or Kruskal–Wallis test. Values sharing a letter in one row are significantly different (*p* < 0.05) in the RIR Tukey test or multiple-comparisons test.

**Table 3 molecules-29-01942-t003:** Content of elements (per kg of wet matter) in spleens of rats from revealed clusters.

	Cl1	Cl2	Cl3	*p*-Value
	Macroelements			
K (mg/kg)	4919 ± 417	5152 ± 229 ^a^	4849 ± 724 ^a^	0.0255
Mg (mg/kg)	246.2 ± 22.0	245.9 ± 23.9	243.4 ± 35.3	0.9160
Na (mg/kg)	23.20 ± 10.86	16.75 ± 9.26	55.19 ± 142.96	0.1345
Ca (mg/kg)	168.0 ± 32.1 ^a^	178.5 ± 35.5 ^b^	133.6 ± 43.2 ^a,b^	<0.0001
	Microelements			
Fe (mg/kg)	3732 ± 1053 ^a^	3376 ± 691 ^b^	2586 ± 1263 ^a,b^	0.0002
Zn (mg/kg)	21.75 ± 1.72	23.91 ± 5.14 ^a^	20.93 ± 2.56 ^a^	0.0061
Cu (mg/kg)	3.17 ± 0.34 ^a^	3.20 ± 0.60 ^b^	2.26 ± 0.48 ^a,b^	0.0000
Mn (mg/kg)	0.56 ± 0.07 ^a^	0.58 ± 0.09 ^b^	0.39 ± 0.12 ^a,b^	<0.0001
Se (mg/kg)	0.62 ± 0.07 ^a,b^	0.71 ± 0.07 ^a,c^	0.48 ± 0.08 ^b,c^	<0.0001
Co (mg/kg)	0.011 ± 0.002 ^a^	0.059 ± 0.013 ^a,b^	0.016 ± 0.014 ^b^	<0.0001
Cr (mg/kg)	0.44 ± 0.06 ^a^	0.48 ± 0.11 ^b^	0.31 ± 0.08 ^a,b^	<0.0001
Ni (mg/kg)	0.04 ± 0.02 ^a^	0.16 ± 0.10 ^a,b^	0.05 ± 0.04 ^b^	<0.0001
Al (mg/kg)	57.93 ± 7.00 ^a,b^	46.68 ± 8.92 ^a^	49.00 ± 12.93 ^b^	0.0002
Sr (mg/kg)	0.29 ± 0.04 ^a,b^	0.19 ± 0.03 ^a^	0.18 ± 0.06 ^b^	<0.0001
Pb (mg/kg)	0.15 ± 0.02 ^a,b^	0.17 ± 0.05 ^a,c^	0.10 ± 0.02 ^b,c^	<0.0001
Cd (μg/kg)	3.12 ± 0.70 ^a^	3.36 ± 0.70 ^b^	1.87 ± 0.53 ^a,b^	<0.0001
B (mg/kg)	87.99 ± 10.94 ^a,b^	69.55 ± 12.66 ^a^	74.11 ± 20.41 ^b^	0.0001
Tl (μg/kg)	13.23 ± 2.37 ^a,b^	8.40 ± 2.52 ^a^	7.93 ± 2.82 ^b^	<0.0001

Data are shown as mean values ± standard deviation (SD). *p*-value ≤ 0.05—significant differences among groups in one-way ANOVA. Values sharing a letter in one row are significantly different (*p* < 0.05) in RIR Tukey test.

**Table 4 molecules-29-01942-t004:** Loadings, eigenvalues, and variances of the significant principal components.

Variables	PC1	PC2	PC3
K	−0.278	−0.249	**0.624**
Mg	−0.080	0.052	**0.751**
Na	0.279	−0.029	−0.408
Ca	−0.682	0.186	0.144
Fe	−0.517	0.338	0.042
Zn	−0.348	−0.028	**0.751**
Cu	**−0.702**	−0.178	−0.170
Mn	**−0.818**	−0.103	−0.162
Se	**−0.840**	−0.306	−0.171
Co	−0.550	**−0.709**	0.002
Cr	**−0.824**	−0.104	−0.293
Ni	−0.435	−0.379	0.315
Al	−0.293	**0.841**	0.321
Sr	−0.429	**0.660**	−0.353
Pb	**−0.785**	−0.165	0.016
Cd	**−0.861**	−0.059	−0.157
B	−0.292	**0.849**	0.252
Tl	−0.433	**0.636**	−0.288
Eigenvalue	11.654		
Variance (%)	33.12	18.40	13.22
Cumulative (%)	64.7		

The most significant loadings are boldfaced.

**Table 5 molecules-29-01942-t005:** Coefficients and average values of canonical variables included in the final model.

	Coefficients of Canonical Variables						
	DF1	DF2	DF3	DF4	DF5	DF6	DF7
Variables	93.63%	2.40%	1.76%	1.16%	0.59%	0.36%	0.08%
Co	6.96363	1.09279	0.128744	0.21518	0.23443	0.147156	0.88032
Sr	−1.04424	−0.29692	0.634247	−0.28200	−0.37835	−0.702481	1.35566
Mn	−0.93829	0.55813	−0.890473	0.24603	−0.52714	0.020905	−1.34120
Ni	0.40805	0.19431	0.391771	−0.99295	0.69071	−0.294469	0.41979
Ca	0.70833	−1.38438	−0.594031	−0.14685	−0.87512	0.201752	0.05789
B	−0.00917	2.01984	−0.940549	1.15620	−1.14442	0.269535	1.01068
Cr	−0.16064	−0.54411	1.663446	0.62522	0.19558	−0.226185	−0.76844
Fe	0.04448	−0.33491	−0.557203	−0.84495	0.07122	−0.695610	−0.56046
Cu	−0.12231	0.25923	−0.959160	−0.87929	−0.10049	1.019662	−0.23898
Na	0.22432	−0.59006	0.533600	0.10718	−0.00830	0.232868	−0.08378
Mg	0.21959	−0.74660	0.613614	0.37265	1.15938	−0.062023	0.00759
Tl	−0.16485	−0.25790	−0.142177	0.60750	0.79269	−0.904604	0.74983
K	0.14740	0.16593	−0.606916	0.08094	0.09168	−0.231532	0.32261
Cd	−0.00744	−0.65129	0.197402	0.67524	0.65959	0.789825	0.55133
Zn	−0.16766	0.00680	−0.084258	0.22020	−0.88783	−0.739514	0.33817
Al	−0.87280	−0.09641	0.495011	−1.72349	1.33040	1.441113	−1.30752
	**Average Values of Canonical Variables**						
CON	6.33793	−0.75139	−0.01459	−0.53807	1.250980	0.412474	0.040574
M	5.42223	1.80042	−1.07040	−1.20665	−0.517138	0.096160	−0.017901
G	7.95468	−0.21525	0.46986	0.54690	−0.207331	−0.634582	0.385911
GM	8.61790	−0.55439	0.44453	0.95220	−0.471548	0.216111	−0.379717
CONplus	−8.32973	−1.72443	−0.71386	−0.03044	−0.658549	0.593416	0.203602
Mplus	−7.50768	−0.87100	−0.70078	−0.40368	0.233689	−0.844036	−0.233464
Gplus	−8.18277	0.66246	2.88315	−0.90017	−0.162521	0.089866	−0.024297
GMplus	−8.43741	1.88802	−0.43238	1.39343	0.461853	0.163634	0.037266

**Table 6 molecules-29-01942-t006:** Classification results of LDA presenting percentage of predicted group membership for actual groups.

	Predicted Group Membership								
Actual Group	Correct Classification	CON	M	G	GM	CONplus	Mplus	Gplus	GMplus
CON	83.33%	10	0	2	0	0	0	0	0
M	100.00%	0	12	0	0	0	0	0	0
G	58.33%	1	0	7	4	0	0	0	0
GM	83.33%	0	0	2	10	0	0	0	0
CONplus	90.91%	0	0	0	0	10	1	0	0
Mplus	91.67%	0	0	0	0	0	11	0	1
Gplus	75.00%	0	0	0	0	1	1	6	0
GMplus	90.91%	0	0	0	0	0	1	0	10
**Total**	**84.44%**	**11**	**12**	**11**	**14**	**11**	**14**	**6**	**11**

## Data Availability

Dataset available on request from the authors.

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
