# Peer review of "Splenic Elemental Composition of Breast Cancer-Suffering Rats Supplemented with Pomegranate Seed Oil and Bitter Melon Extract"

_molecules, 2024, doi:10.3390/molecules29091942_

Round 1
Reviewer 1 Report
Comments and Suggestions for Authors
The manuscript, “Splenic elemental composition of breast cancer suffering rats supplemented with pomegranate seed oil (PSO) and bitter melon extract (BME)” to investigate the influence of diet modifications with PSO and BME on the content of selected elements in the spleen of rats in physiological state and co-existing mammary tumorigenesis
1. In my opinion, The introduction is descriptive but lacks details regarding pomegranate seed oil and bitter melon extract, even though the author has included references to previous studies. However, the article should contain sufficient information about the extracts and their active substances, including previous research on how these active substances affect splenic elemental composition and breast cancer.
2. The dosages of pomegranate seed oil and bitter melon extract utilized in the experiment are referenced
3. Line 288-290; “As discussed previously there were no spontaneous tumors of any kind in animals untreated with DMBA whereas mammary cancer incidence in ’plus’ groups was 42% in CONplus, 67% in Mplus, and 100% in Gplus and GMplus groups, respectively” In which section of the manuscript does the author engage in the discussion? Furthermore, there are no results from cancer incidence presented in the manuscript."
4. The discussion provides a thorough explanation of each element of the mechanisms affecting breast cancer.
5. Line 448; After 10 min
Comments on the Quality of English LanguageAuthor Response
Please see the attachment.

Reviewer 2 Report
Comments and Suggestions for Authors
The manuscript presents another study conducted by the research teams using pomegranate seed oil and bitter melon extract. The research team has already published several papers on this topic.
The present research investigates the changes that appeared in the elemental composition of rats' spleen. The rats were fed with pomegranate seed oil (PSO) and/or bitter melon extract and then subjected to a chemical carcinogen in order to develop mammary tumors. The results are clearly presented. However, there are several improvements which can be made:
Title – in my opinion, it is not necessary to add the abbreviations (PSO, BME) in the title.
Introduction – the introduction section is too long and too exhaustive.
There is no need to include information about its structure, the role of the white and red pulp, or how the spleen is perfused (lines 56-61). Also, I recommend excluding the text containing the impaired spleen functions which are not related to cancer.
Lines 119-122 – can be transferred to the discussion section
Lines 126-128 – please reformulate: “Therefore, the aim of the present study was to.....”
Material and methods:
Line 448: After 10 minutes, « the » fresh extract was filtered,
- Why the BME was not administrated as PSO, in a fixed volume per day, by gavage?
Line 470: After 1 week of adaptation «, » animals were randomly divided into eight groups…
Line 484: the explanations about the "plus" groups should be presented immediately after the characteristics of experimental groups (line 480).
Tables 7 and 8 should not be included in the manuscript, but rather introduced as supplementary material.
Line 538 - To eliminate the outliers «, the » obtained results were…
Results:
Line 134 - In dietary supplements « the » levels of…
Lines 138-139 « …all determined elements, except Se and Cd, were determined. » - please reformulate
Lines 139–143: Taking into account that for PSO, most macro and microelements were below LOD, and K was 13.66 compared to 18570, Fe was 4.13 compared to 225.9 from fodder) and also for Bitter melon (dry or fluid) are considerable differences compared to the elemental composition of Labofeed H, I would refrain from stating that: modification with PSO or/and BME to a small extent influenced the mean dietary intake of investigated elements in all experimental groups.
Also, in the Results section, it is advised to include results only, and not personal observations.
Discussion:
The discussion chapter is again, too long. Please be more concise and try to discuss more your own results.
Lines 265 – 287 – can be condensed to 1-2 phrases.
Lines 320-335 – please condense
Lines 350 – 395 – please condense
Also, please reformulate the Discussion and Conclusion part so clearer ideas can be drawn.
Comments on the Quality of English LanguageThe English language is fine, but please revise the manuscript as there are some missing commas.
Round 2
Reviewer 1 Report
Comments and Suggestions for Authors
1. The LOD and LOQ values that expressed in Table S1 were quite similary. How do you calculate LOD and LOQ? Additionally, the method validation required follows guidelines such as ICH, EURACHEM, etc. Please explain further.
2. Why the tested animal received 0.15 mL of pomegranate seed oil (PSO) each animal rather than per kilogram of body weight. Please provide more details.
3. It is necessary to provide appropriate information about PSO active compounds, such as tocotrienols and tocopherols, to Table S2.
4. Table S1 showed that the total phenolic content and the chlorogenic acid level of the BME 1% aqueous extract is 640 g/L and 6.33 mg/100 g dw, respectively. Please give further details about the other flavonoid or phenolic chemicals that have been documented in BME.
Author Response
Reviewer #1
We are greatly obliged for having received the Reviewer’s opinions and comments on our manuscript. We are very grateful for the time and effort they spared for revision of our article. We have read thoroughly all valuable comments, advice and suggestions and we found them really inspiring and helpful. The replies to specific comments are listed below. All changes in the manuscript are marked in pink. We do hope you will find them suitable and sufficient.
1. The LOD and LOQ values that expressed in Table S1 were quite similary. How do you calculate LOD and LOQ? Additionally, the method validation required follows guidelines such as ICH, EURACHEM, etc. Please explain further. Response: Thank you for pointing out the issue of method validation. We added the description of the formula used for LOD and LOQ calculations between lines 474-479. There was: Limits of detection (LOD) and limit of quantitation (LOQ) were calculated for each element according to the IUPAC recommendation [54] and are presented in Table S1. There is: Limits of detection (LOD) and limit of quantitation (LOQ) were calculated for each element by summing the mean of ten measurements of blank samples and three standard deviations for LOD, or ten standard deviations for LOQ, following the IUPAC recommendation [54]. These values are presented in Table S1. Blank samples were prepared under the same conditions as the samples. We would also like to add the information that the analytical procedure used single weights of test samples (which has been added to the text in line 464): Due to the limited amount of material, the analytical procedure was conducted for a single weight of test samples. The reason for that was the limited amount of tissues and the will to provide as much available material for analysis as possible. Despite the use of CRM to verify the analytical procedure (mentioned in text lines 450-453), there are obvious differences in homogeneity compared to the natural tissue. Even if we tried to replicate natural conditions (similar weights of CRM and test samples) there are limitations associated with reproducing the exact homogeneity. The analytical procedure was developed by accepted standards and analytical guidelines, such as Eurachem. The procedure was aligned with the laboratory's accreditation standards (accreditation scope AB 1525 by Polish Centre for Accreditation for metals in food by ICP-MS) and based on IUPAC recomendations (as cited ref. 53) and earlier application of the procedure (as cited ref. 21). The use of CRM (mentioned lines 450-453, 455-457) and careful monitoring of validity of the results were integral parts of the analysis. Although full validation documentation is not presented in "4.3 ICP-MS elements determination", we would like to ensure that our method provides reliable and accurate results.
2. Why the tested animal received 0.15 mL of pomegranate seed oil (PSO) each animal rather than per kilogram of body weight. Please provide more details. Response: We would like to explain, that dosages of both supplements were based on previously published papers and previous experiments, especially our previous experiment comparing the CLA and PSO as a source of CLnA (BiaÅ‚ek et al. Prostaglandins and Other Lipid Mediators 131 (2017) 9–16, 10.1016/j.prostaglandins.2017.05.004). It is a common practice in such experiments to give one, strictly fixed dose of experimental dietary factors for the whole time period of the experiment, even if the animals are still growing and their body mass changes. The animal daily PSO intake, used in present experiment, was also intended to reflect the way of using dietary supplements by humans. It is recommended to use one strictly fixed dose, recommended by the manufacturer, and not to adjust the dose of supplements according to their body
weight, but use the recommended amount of gelatin capsules containing oily filling. Moreover, taking into account the fact that animals were weighted weekly, each animals had slightly different body mass and their mass changed during experiment, it was impossible to change the dose for each animal each day. Another practical aspect is the fact, that there are some restriction regarding the volume given via gavage, which we have mentioned before. We would like to mention also that to our best knowledge we did not come across any studies on model animals with such an attempt suggested by the Reviewer (adjusting dose of experimental dietary factor to single animal based on its body mass each day). We do hope that such explanation will be sufficient.
3. It is necessary to provide appropriate information about PSO active compounds, such as tocotrienols and tocopherols, to Table S2. Response: As requested, detailed information about the content of tocopherols in PSO was added to Table S2.
4. Table S1 showed that the total phenolic content and the chlorogenic acid level of the BME 1% aqueous extract is 640 g/L and 6.33 mg/100 g dw, respectively. Please give further details about the other flavonoid or phenolic chemicals that have been documented in BME. Response: We would like to explain that the BME used in present study, as well as the dry BM fruits (commercially available tea) used for preparation of BME, were thoroughly analyzed by the Department of Pharmacognosy and the Department of Food Chemistry and Nutrition, Medical College, Jagiellonian University. Our collaborators are the experts in the field of dietary supplements and medicines of natural origin. They analyzed BM dry matter and BME extract regarding the content of active substances and only those results, presented in our manuscript, have been obtained. We would like to emphasize, that all results of undertaken analyzes were published previously in our paper (Białek et al. 2019, DOI: 10.1002/ejlt.201800420) and now, at the specific request of Editorial Office, were included in Table S2 of present manuscript. Unfortunately, we cannot include any new data regarding BME composition.

Reviewer 2 Report
Comments and Suggestions for Authors
I have no further comments. The papers can be published as it is.
Author Response
Reviewer #2
1. I have no further comments. The papers can be published as it is.
Response: We are greatly obliged for having received the Reviewer’ opinions and comments on our manuscript. We are very grateful for the time and effort they spared for revision of our article. Thanks to all valuable comments, advice and suggestions our manuscript was significantly improved.
